# The Bonar Score in the Histopathological Assessment of Tendinopathy and Its Clinical Relevance—A Systematic Review

**DOI:** 10.3390/medicina57040367

**Published:** 2021-04-09

**Authors:** Maria Zabrzyńska, Dariusz Grzanka, Wioletta Zielińska, Łukasz Jaworski, Przemysław Pękala, Maciej Gagat

**Affiliations:** 1Faculty of Medicine, Collegium Medicum in Bydgoszcz, Nicolaus Copernicus University in Torun, 87-100 Toruń, Poland; 2Department of Pathology, Faculty of Medicine, Collegium Medicum in Bydgoszcz, Nicolaus Copernicus University in Torun, 87-100 Toruń, Poland; d_grzanka@cm.umk.pl; 3Department of Histology and Embryology, Faculty of Medicine, Collegium Medicum in Bydgoszcz, Nicolaus Copernicus University in Torun, 87-100 Toruń, Poland; w.zielinska@cm.umk.pl (W.Z.); mgagat@cm.umk.pl (M.G.); 4Private Center of Surgery, 87-100 Torun, Poland; lukaszmjaworski@gmail.com; 5International Evidence-Based Anatomy Working Group, Department of Anatomy, Medical College, Jagiellonian University, 30-034 Krakow, Poland; pekala.pa@gmail.com; 6Faculty of Medicine and Health Sciences, Andrzej Frycz Modrzewski Krakow University, 30-034 Krakow, Poland; 7Lesser Poland Orthopedic and Rehabilitation Hospital, 30-034 Krakow, Poland

**Keywords:** Bonar score, tendon, tendinopathy, microscopy, tendon pathology

## Abstract

This study aimed to perform a comprehensive systematic review, which reports the role of the Bonar score in the histopathological assessment of tendinopathy and its clinical relevance. To identify all of the studies that reported relevant information on the Bonar scoring system and tendinopathy, an extensive search of the major and the most significant electronic databases (PubMed, Cochrane Central, ScienceDirect, SciELO, Web of Science) was performed. A systematic review of the literature was conducted according to the Preferred Reporting Items for Systematic Reviews and Meta-Analyses (PRISMA) guidelines. The extracted data included—year of study, geographical location, type of the study, radiological modifications, gender, number of patients, region of tendinopathy, mean age, control group, characteristics of the Bonar score and alterations in the scale, mean Bonar score, number of investigators, area of tendon investigation, clinical and radiological implications. An extensive search of the databases and other sources yielded a total of 807 articles. Eighteen papers were finally included in this systematic review, and of these, 13 original papers included the clinical and radiological implications of tendinopathy. Radiological evaluation was present in eight studies (both magnetic resonance imaging (MRI) and ultrasound (US)). The clinical implications were more frequent and present in 10 studies. Using the Bonar score, it is easy to quantify the pathological changes in tendinous tissue. However, its connection with clinical and radiological evaluation is much more complicated. Based on the current state of knowledge, we concluded that the neovascularization variable in the Bonar system should be reconsidered. Ideally, the microscopic assessment score should follow the established classification scale with the radiological and clinical agreement and should have a prognostic value.

## 1. Introduction

Chronic tendon pathology is the main problem in competitive and recreational athletes (tendon disorders lead to about 30% of orthopedic consultations) [1,2,3]. Achilles tendinopathy occurs most frequently in runners, basketball, volleyball and football players [4,5]. On the other hand, tibialis posterior tendinopathy is most common in runners. Hamstring tendon disorders are often found in sprinters, jumpers and football players. Regarding upper limb tendinopathies, arm extensor and flexor tendon disorders are usually present among tennis, golf players and throwing athletes, specifically javelin throwers and baseball players, as well as triceps tendon pathology [4,5,6]. Rotator cuff and biceps tendinopathy are most commonly encountered in swimmers, American football players and javelin throwers [7]. Although tendons seem to be relatively simple structures, made of connective tissue, recent studies revealed that they are characterized by complex biology and impaired regeneration processes [8,9,10,11]. Despite the enormous progress in therapeutic options in tendinopathy treatment, the etiopathology and the histopathology of tendon disorders are yet not completely understood [10,12,13,14]. This knowledge is especially important for the more effective management of tendon injuries, both acute and chronic. When compared to bone and skeletal muscles, the recovery process of injured tendons is relatively slow [15,16,17]. This phenomenon is associated with reduced metabolism of tenocytes, which are characterized by 7.5 times lower oxygen consumption than skeletal muscles [18]. This allows the tendon to resist long-term stress. On the other hand, the low metabolic rate may impair the tissue regeneration process and increase the risk of tendinous tissue degeneration [13]. The abundant proliferation of tenocytes, collagen composition and architecture alterations, ground substance accumulation, and chaotic neovascularization are the characteristics of the tendon degenerative process, which clinically is recognized as tendinopathy.

Currently, only a few studies quantifying the histopathological alterations in tendinopathy using reproducible scales such as Movin or Bonar systems are available [19,20,21]. The most popular and established scale to assess the microscopic alterations of tendons is the Bonar score. This semiquantitative scoring system evaluates four main variables—tenocyte morphology alterations, the ground substance accumulation, the extent of neovascularization, and collagen bundle architecture [20]. For each variable, zero indicates normal parameters and three a markedly abnormal appearance. This yields a total of zero (normal tendon) to 12 points (the most severe detectable pathology). Additionally, numerous authors tried to modify this system by adding new variables, such as the number of inflammatory cells, the number of tenocytes/cellularity, the presence of calcifications, and the alterations in tenocytes’ cytoplasm [19,21]. Moreover, the investigation area of the tendon specimen is not clearly defined. Some authors assess the whole sample of tendon, while others decide to evaluate only the most degenerated region. The number of investigating pathologists is also a moot point in the literature as none of the authors defined a necessary number of experts to provide a reliable verdict. Fearon et al. revealed that the same tendon sections scored on separate days by two examiners using the original Bonar score showed poor inter-tester reliability of *r*^2^ = 0.32. However, after some modifications it increased to *r*^2^ = 0.71 [21]. Moreover, staining procedures also vary in different studies. Furthermore, the prognostic value and clinical significance of the Bonar score have not been yet established in the literature.

This study aimed to perform a comprehensive systematic review, which reports the role of the Bonar score in the histopathological assessment of tendinopathy and its clinical relevance.

## 2. Materials and Methods

### 2.1. Search Strategy

To identify all of the studies that reported relevant information on the Bonar scoring system and tendinopathy, an extensive search of the major and the most significant electronic databases (PubMed, Cochrane Central, ScienceDirect, SciELO, Web of Science) was performed. A systematic search was conducted in October 2020. The database investigation was done using combinations of the following key terms—Bonar AND (tendon OR tendinopathy OR score), with no limits regarding the year of publication. An additional intensive search through the references of all identified studies was conducted. A systematic review of the literature was carried out according to the Preferred Reporting Items for Systematic Reviews and Meta-Analyses (PRISMA) guidelines (Figure 1) and Appendix A.

### 2.2. Eligibility Assessment

The search was done independently by two authors (M.G. and M.Z.). Three independent reviewers (M.G., M.Z., Ł.J.) screened all of the articles identified for a title, abstract, and full text concerning the application of the Bonar score in tendon pathology. All preclinical and clinical studies (in vivo and in vitro) in English were evaluated and analyzed in this systematic review. Studies assessing the histopathology of tendons with the Bonar scoring system have been considered eligible for inclusion. We searched for the Bonar score application in tendon pathology of various studies of tendinopathy. We also focused on the clinical implications of the microscopic alterations in tendinous tissue. Non–English language studies, case studies, reviews, letters to editors, conference abstracts, or studies containing incomplete or irrelevant data and studies of level V evidence were excluded from the analysis. Exclusion criteria also included any Bonar score application in structures other than the tendons, e.g., menisci. Moreover, studies in which the Bonar score was used for purposes other than pathology assessment were not taken into consideration. The animal studies were also excluded. The senior author (P.A.P.) made the final decision on article inclusion if there was disagreement among authors.

### 2.3. Data Extraction

Three independent reviewers (M.G., M.Z., Ł.J.) extracted the relevant data including the year of study, geographical location, type of the study (cadaveric, human, in vitro), radiological modifications, gender, number of patients, region of tendinopathy, mean age, control group, characteristics of the Bonar score and alterations in scale, mean Bonar score, number of investigators, area of tendon investigation, and clinical and radiological implications.

### 2.4. Risk of Bias Assessment

We also supported this systematic review with the AQUA Tool, which judges the quality of an anatomical study by assessing risks of bias due to the methods and the reporting of results [22]. The tool evaluates the risks of bias in an anatomical study from the perspective of five key domains—(1) Aim and subject characteristics, (2) Study design, (3) Characterization of methods, (4) Descriptive anatomy, and (5) Results reporting (Figure 2).

## 3. Results

An extensive search of the databases and other sources yielded a total of 807 articles. Duplicates and records not meeting qualification criteria were rejected. The 57 full-text original articles were assessed by authors for potential eligibility. Sixteen of the investigated articles were finally included in this systematic review (Figure 1). The risk of eventual bias according to the AQUA protocol was summarized in Table 1 and Figure 2.

### 3.1. Characteristics of Included Studies and Demographic Data

The characteristics and demographic data of the included articles are presented in Table 2.

This systematic review evaluates a total of 16 papers (*n* = 552 subjects). The studies were published between 2004 and 2019. The included studies originated from variable geographical regions such as Asia, Europe, North America, and Australia. The mean age of specimens ranged from 22 to 67.2 years. The supraspinatus tendon dominated among the tendon samples (*n* = 5), while the rarest were hamstring tendons (*n* = 1) and tibialis posterior tendon (*n* = 1). The control groups were rarely included (*n* = 8) and mainly consisted of semitendinosus and gracilis tendons (*n* = 3). The mean Bonar score in the pathological tendons group ranged from 3.2/12 to 9.53/12 and in the modifications group, it varied between 11.6 and 14.4/20 points. In control groups, the value reached from 0/12 to 7.8/12 points in the classical Bonar system, while for the modified scale, it was between 9/20 and10.43/20 points.

### 3.2. Bonar Score and Its Modifications

The authors of included studies used mostly the four-variable classical Bonar score [19,20,23,24,25,26,29,31,32] (Table 3).

However, a comparable number of authors modified the classical Bonar score to improve the diagnostic and microscopic evaluation [21,27,28,30,33,34,35]. The changes mainly concerned local fibroblast counting (manually or using dedicated software), cytoplasm morphology alterations, isolated vascular assessment using the criteria taken from the Bonar score, evaluation of calcifications as well as adipocytes. In four studies, there were two investigators, in the other five, just one investigator, while in four studies, three investigators evaluated specimens. However, in three studies, the information about the number of experts was not available. Regarding the staining methods, the most frequent was Alcian Blue (nine studies). Other methods, such as Mallory, Masson Trichrome, and additional immunohistochemical techniques, were rather rare. In seven studies, authors did not use any special staining method, except the hematoxylin and eosin staining. Some of the authors augmented the microscopic evaluation with immunohistochemical reactions, e.g., for apoptosis [27], collagen I type [24], or the neovascularization process detection [28]. The area of investigation of the tendon sample was also an issue as in the majority of studies, authors did not indicate the evaluated region. Three authors assessed the most severely degenerated region of specimen [20,21,28], while one author analyzed the whole sample [24]. Only one author evaluated randomly selected slides of tissue samples [31].

### 3.3. Clinical and Radiological Implications

Authors of 13 original papers introduced clinical and radiological implications to their studies (Table 4).

Radiological evaluation was usually based on magnetic resonance imaging (MRI) (four studies) and ultrasound (US) (four studies). All MRI-involving studies revealed no correlation between histopathological and radiological alterations [23,24,26,34]. Only one US study by Zabrzynski et al. revealed the association between the echogenicity of biceps tendon disturbance and the degenerated area of the tendon. However, the authors did not specify numerical data and statistics. On the other hand, the clinical implications in histopathological studies were more frequent and present in nine papers. Clinical tests dedicated for Tibialis posterior tendon (PTT), long head of biceps tendon (LHBT) and supraspinatus tendon (SST) showed no correlation with histopathological changes in tendinous tissue [23,24,28]. Moreover, the pain scale assessment (VAS) score evaluation revealed no correlation with microscopic alterations [28]. On the contrary, Fearon et al. found a correlation between greater trochanteric pain syndrome and higher Bonar score [35]. In turn, Lundgreen et al. compared SST in the smoking and non-smoking populations. They showed that smokers presented significantly more advanced degenerative changes with abundant apoptotic features in cells with the reduction of the local fibroblast population [30]. Lundgreen et al. presented that macroscopically torn SST demonstrated a significant degeneration of their structure compared to intact samples [27]. Contrary to these findings, Sethi et al. showed that macroscopic investigation did not correlate with Bonar score in light microscopy [24]. What is more, Kurdziel et al. observed that in torn and intact RC populations, the LHBT microscopic alterations measured in the Bonar score showed no statistically significant differences [31]. Cook et al. in the patellar tendon study of a population, which underwent Anterior cruciate ligament (ACL) reconstruction using a bone–tendon–bone autograft, revealed pathological alterations in tendinous tissue. However, there were no differences between subjects with and without pathological alterations in tendons in recovery after the operation, training, and anthropometric measures [32]. Okazaki et al. revealed that mechanical stimuli, such as scratching to the hamstring tendon at the time of the preparation of the graft during the ACL reconstruction surgery, induced the degeneration in tissue, disturbance in collagen I type architecture and composition, together with impaired Bonar score [33].

## 4. Discussion

Evaluation using two scoring systems is commonly used in tendon pathology. It includes the Movin score and the Bonar score, which is more widespread [20]. The variables in the Movin scale are fiber structure, fiber arrangement, rounding of the nuclei, regional variations in cellularity, increased vascularity, decreased collagen stainability, hyalinization, and glycosaminoglycans content. However, the variables included in the Bonar scale are easier and faster to assess. Moreover, Maffulli et al. showed that Movin’s and Bonar’s scores have a high correlation and assess similar characteristics and variables of tendon abnormalities (*p* < 0.0001) [20]. The variable of hyalinization is the main factor that differs between these two scoring systems (it is not present in the Bonar scale). The hyalinization process rarely occurs in tendinopathy. Thus, Maffulli et al. suggested removing this variable from the Movin scoring system. Another difference is cellularity, included in the Movin score and absent in the Bonar score. However, the authors of some of the included studies additionally supported their findings by introducing the fibroblast counting as a fifth variable into the modified Bonar score [21,33,34]. This upgrade should be reconsidered in further studies, as it is known that the apoptosis process present in tendinopathy tends to alter the population of fibroblasts [36,37]. The issue of inflammation and inflammatory cells was examined in a few studies. Fearon et al. and Zabrzynski et al. evaluated the inflammatory infiltration in tendinous tissue. However, in both studies, the number of such cells was very modest [19,35].

Regarding additional staining methods, numerous authors applied Alcian Blue, Masson Trichrome, Mallory, and others. These methods are dedicated to visualizing the non-collagenous extracellular matrix. For practical purposes, these techniques generate extra costs, are time-consuming, and are not necessary to differentiate the collagen from the non-collagenous extracellular matrix [20]. Moreover, some of the authors used immunohistochemical (IHC) techniques to improve visualization of the vascular bed, using CD31 and CD34 [28], type I collagen [33] or apoptosis processes [27]. These additional methods are not incorporated into scoring systems but usually help to assess the pathology in more detail.

The number of tissue sample investigators also varies a lot between the studies. However, it is generally suggested that two independent observers or one experienced musculoskeletal expert ensure the objective assessment of slides [21]. The same problem was observed considering the area of microscopic investigation. Still, there are no clear instructions regarding the evaluated area. Thus, several approaches may be applied, such as assessing the most severely changed region, randomly selected fields, or the entire specimen. However, as the degeneration of tendinous tissue usually occupies the whole tendon specimen, probably the entire slide should be taken into account in the scoring system.

In 8/18 studies included in this systematic review, the authors did not introduce the control group. In other cases, the semitendinosus and the gracilis tendon (STG) specimens were the most frequent control group. The alterations in the course of tendinopathy in all tendons of the upper and lower limbs are usually similar and include disrupted collagen architecture, tenocyte morphological changes, neovascularization process, and expansion of the ground substance. The control group seems to be unnecessary in chronic tendinopathy due to well-known and established pathological pattern.

None of the included sonographic modality studies presented a clear radiological association with histopathology. Authors of the sonographic study of the LHBT showed abundant histopathological alterations and concomitant sonographic changes, such as hypoechogenicity and increased tendon diameter, however, no statistically significant correlation was found [25]. Cook et al. suggested that imaging abnormalities such as hypoechogenicity requires a high accumulation of the ground substance [32]. Moreover, none of the included studies showed an association between microscopic findings and the MRI modality. Considering the clinical implications, Lundgreen et al. revealed that the severely torn tendons had a higher Bonar score, compared with an intact tendon controlled macroscopically [27]. Furthermore, mechanical stimuli, such as scratching of the tendon, resulted in a higher Bonar score in STG [33]. These two studies presented that the more mechanically injured the tendon is, the worse the histopathological alterations that are observed. Moreover, Lundgreen et al. observed excessive apoptosis of fibroblasts in the smoking population, which correlated with a higher Bonar score [30]. Pain in the area of the greater trochanter also correlated with a higher Bonar score in obtained gluteal tendons [35]. The previously mentioned studies suggest that the Bonar scoring system had better agreement with clinical data than with radiological modalities, however this topic should be further explored.

The neovascularization issue in tendinopathy is ambiguous. On the one hand, the new vessel formation is a form of tissue regeneration and healing. But on the other hand, the neovascularization process is well documented in advanced tendinopathies and considered a pathological phenomenon [38,39,40]. Some authors suggested that the new vessel expansion may be connected with the mediation of pain [38,39]. Newly formed vessels are followed by nerves and may contain increased levels of calcitonin gene-related peptide (CGRP) and substance P responsible for pain sensation [41]. Thus, chronic tendinopathy is described as a painful process with a chaotic vascular bed expansion, which is an effect of impaired tendon regeneration. However, Zabrzynski et al. did not find the correlation between pain scale assessment (VAS) and the neovascularization process evaluated by adopting the criteria from the Bonar scoring system [28]. Furthermore, Singaraju et al. assessed substance P and CGRP in nerve endings using IHC and did not find an association between pain and histopathology [42]. The neovascularization process is regulated by the local microenvironment—ECM and mediators [43,44,45,46]. Zabrzynski et al. showed that chronic tobacco smoking resulted in reduced formation of new capillaries and connected this fact with very dense non-collagenous ECM [47,48]. Moreover, the authors observed that in the non-smoking population, the pathological tendinous tissue was definitely looser. Contrary to this, the Bonar score in smokers was lower due to decreased neovascularization. Lundgreen et al. also observed the negative influence of tobacco smoking on tendon histopathology and the SST [49]. Specimens from the smoker group presented significantly more advanced degenerative changes. Moreover, the authors noted the apoptotic features in the smoking population samples [30]. Apparently, the Bonar score, well-established in tendon pathology, may be biased by a well-known factor, such as tobacco smoking. Moreover, a complete lack of vascularity was assumed by Fearon et al. in their study as a pathology, equally with the abundant neovascularization process [21]. Thus, the neovascularization variable should be reconsidered in the Bonar score as well as the occurrence of avascular regions.

This study was limited in several ways. The main problem was heterogeneity in the methodology of the included studies (authors own modifications of the Bonar scoring system, number of investigators, type of tendinous tissue material, staining methods, the selected area of microscopic examination, and control group inclusion). Moreover, in some studies, there was no detailed description of patients’ characteristics such as comorbidities, tobacco use, etc., which may strongly influence the results. The modifications of the original Bonar scale also altered the obtained scores. As a result, the total score was different in selected papers, which may have an impact on its relation with clinics. The AQUA Tool revealed the high risk of bias in the collected papers in the issues of subject’s characteristics (e.g., age, male/female) and results reporting (the Bonar scores were missed or not clearly presented). Okazaki et al. and Cook et al. presented a population with mean age lower than other authors and it could be connected with higher risk of bias. In the case of unclear risk of bias, the main concern was the staining methods, area of specimen investigation as well as the number of investigators. In four studies, there were two investigators, in the other five, just one investigator, while in four studies, three investigators evaluated specimens. However, in three studies, the information about the number of experts was not available.

## 5. Conclusions

In the presented study, we explored the Bonar score application in various tendon pathology and also the modifications of the scoring system introduced by the authors to improve the microscopic assessment. However, only a few authors compared the clinical findings with histopathological alterations. Using the Bonar score, it is easy to quantify the pathological changes in tendinous tissue. However, linking it with clinical and radiological evaluation is much more complicated. In tendinopathy, we should aim to unite the results of microscopic examination with various radiological techniques and clinics. Moreover, the results may be biased by additional factors, such as tobacco smoking. The neovascularization variable in the Bonar system should be reconsidered according to actual knowledge. Ideally, the microscopic assessment score should follow the established classification scale with the radiological and clinical agreement and having a prognostic value.

## Figures and Tables

**Figure 1 medicina-57-00367-f001:**
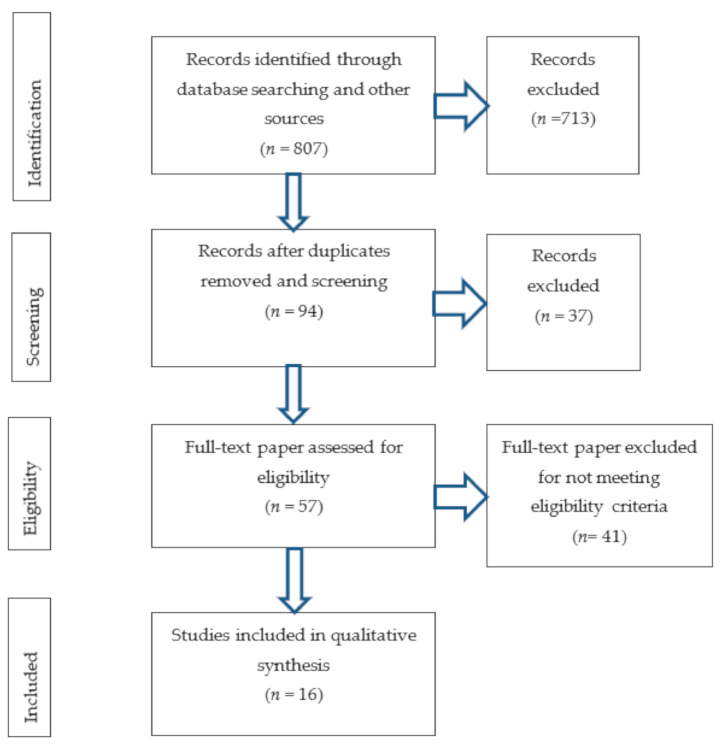
Flow diagram and search strategy.

**Figure 2 medicina-57-00367-f002:**
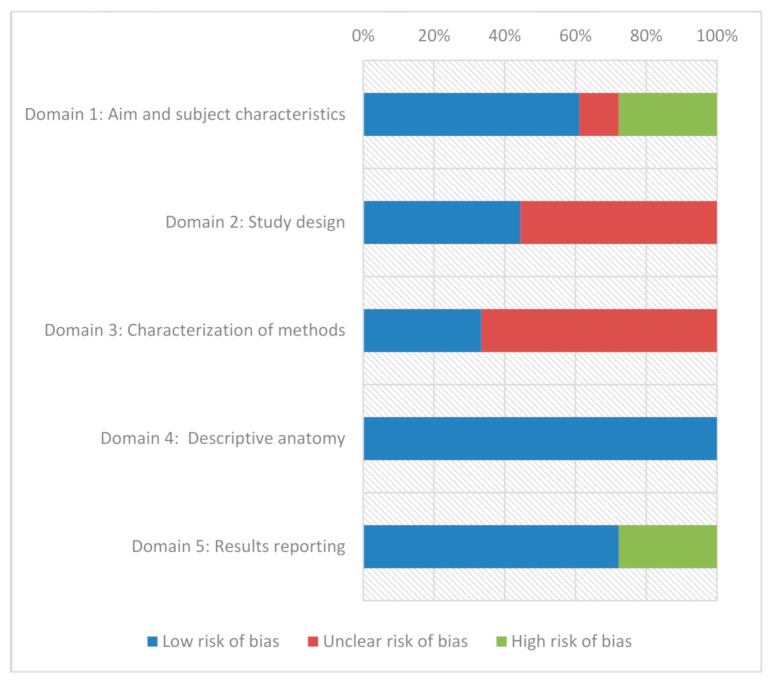
Risk of bias assessment for individual domains using the AQUA tool.

**Table 1 medicina-57-00367-t001:** Risk of bias assessment of each included study using the AQUA tool.

Author	Objective(S) and Subject Characteristics	Study Design	Methodology Characterization	Descriptive Anatomy	Reporting of Results
Maffulli et al. [20]	Risk: Low	Risk: Low	Risk: Low	Risk: Low	Risk: Low
Albano et al. [23]	Risk: Low	Risk: Unclear	Risk: Unclear	Risk: Low	Risk: Low
Fearon et al. [21]	Risk: High	Risk: Unclear	Risk: Low	Risk: Low	Risk: Low
Sethi et al. [24]	Risk: Low	Risk: Unclear	Risk: Low	Risk: Low	Risk: Low
Zabrzynski et al. [25]	Risk: Low	Risk: Unclear	Risk: Unclear	Risk: Low	Risk: High
Nuelle et al. [26]	Risk: Low	Risk: Unclear	Risk: Unclear	Risk: Low	Risk: Low
Lundgreen et al. [27]	Risk: Low	Risk: Low	Risk: Unclear	Risk: Low	Risk: Low
Zabrzynski et al. [28]	Risk: Low	Risk: Unclear	Risk: Low	Risk: Low	Risk: High
Lundgreen et al. [29]	Risk: Low	Risk: Low	Risk: Unclear	Risk: Low	Risk: High
Lundgreen et al. [30]	Risk: Low	Risk: Unclear	Risk: Unclear	Risk: Low	Risk: Low
Kurdziel et al. [31]	Risk: High	Risk: Low	Risk: Low	Risk: Low	Risk: Low
Cook et al. [32]	Risk: Low	Risk: Unclear	Risk: Unclear	Risk: Low	Risk: High
Okazaki et al. [33]	Risk: Low	Risk: Low	Risk: Unclear	Risk: Low	Risk: Low
Docking et al. [34]	Risk: High	Risk: Unclear	Risk: Unclear	Risk: Low	Risk: Low
Zabrzynski et al. [19]	Risk: High	Risk: Low	Risk: Unclear	Risk: Low	Risk: Low
Fearon et al. [35]	Risk: High	Risk: Low	Risk: Low	Risk: Low	Risk: Low

**Table 2 medicina-57-00367-t002:** Included studies of the Bonar score application in various tendons localizations. Demographic data.

Author	Study Type	Level of Evidence (I–IV)	Country	Year of Publication	Region of Pathology	No. of Samples	Age/SD	Control Group	Mean Bonar (0–12)/SD	Mean Bonar Control (0–12)/SD
Maffulli et al. [20]	in vivo	II	Italy/UK	2008	SST ^1^	88	58.2/n/a	SST	9.53/SD = 1.55	1.9/SD = 1.29
Albano et al. [23]	In vivo	IV	Italy	2018	PTT ^2^	19	46/n/a	n/a	8/n/a	n/a/n/a
Fearon et al. [21]	In vivo	III	Australia/Canada	2014	Gluteal tendons	35	n/a/n/a	n/a	14.4/20; 13/20; 11.6/20 /SD = 14.4 (1.50)	n/a/n/a
Sethi et al. [24]	In vivo	III	USA	2018	SST	85	61.6/SD = 9.7	n/a	7.5/SD = 2.7	n/a/n/a
Zabrzynski et al. [25]	In vivo	IV	Poland	2018	LHBT ^3^	19	54/n/a	n/a	8.2/n/a	n/a/n/a
Nuelle et al. [26]	In vivo	IV	USA	2018	LHBT	16	44.25/n/a	n/a	7.9/SD = 1.8	n/a/n/a
Lundgreen et al. [27]	In vivo	III	Norway	2018	SST	19	54/n/a	SSC ^4^	5.5/n/a	1/n/a
Zabrzynski et al. [28]	In vivo	III	Poland	2018	LHBT	32	52/SD = 10.5	STG ^5^	n/a/n/a	n/a/n/a
Lundgreen et al. [29]	In vivo	III	Norway	2011	SST	25	57.7/n/a	SSC	n/a/n/a	n/a/n/a
Lundgreen et al. [30]	In vivo	III	Norway	2014	SST	25	52/SD = 6.4	n/a	13.5/20/SD = 1.375	9/20/SD = 3
Kurdziel et al. [31]	In vivo	III	USA	2015	LHBT	34	67.2/SD = 10.7	LHBT	7.3/SD = 1.6	7.8/SD = 1.2
Cook et al. [32]	In vivo	IV	Australia	2004	Patellar tendon	50	29.0/SD = 9.1	n/a	n/a/n/a	n/a/n/a
Okazaki et al. [33]	In vivo	III	Japan	2019	STG	6	22/n/a	STG	3.2; 5.7; 7.2 SD = 1.5;1.5;0.4	0.8/SD = 0.4
Docking et al. [34]	In vivo	IV	Australia	2018	Gluteal tendons	26	n/a/n/a	n/a	8.12/n/a	0/n/a
Zabrzynski et al. [19]	In vivo	III	Poland	2017	LHBT	39	53/SD = 10.48	STG	8.2/SD = 2	0
Fearon et al. [35]	In vivo	II	Australia	2014	Gluteal tendons	34	n/a/n/a	Gluteal tendons	12.65/20/SD = 2.0	10.43/20/SD = 4.84

^1^ Supraspinatus tendon; ^2^ Tibialis posterior tendon; ^3^ Long head of biceps tendon; ^4^ Subscapularis tendon; ^5^ Semitendinosus and gracilis tendon.

**Table 3 medicina-57-00367-t003:** Classical and modified Bonar score specifications in the literature.

**Classical Bonar Score**				
**Author**	**The Components of Bonar Score**	**Number of Investigators**	**Area of Specimen Investigation**	**Additional Staining Methods**
Maffulli et al. [20]	1. Tenocytes 2. Ground substance 3. Collagen 4. Vascularity	2	The most pathological area	Alcian Blue
Albano et al. [23]	1. Tenocytes 2. Ground substance 3. Collagen 4. Vascularity	1	n/a	n/a
Sethi [24]	1. Tenocytes 2. Ground substance 3. Collagen 4. Vascularity	3	Total area of specimen	Alcian Blue
Zabrzynski et al. [25]	1. Tenocytes 2. Ground substance 3. Collagen 4. Vascularity	1	n/a	n/a
Nuelle et al. [26]	1. Tenocytes 2. Ground substance 3. Collagen 4. Vascularity	n/a	n/a	Masson’s Trichome, Picrosirius Red staining, Verhoeff’s, IHC ^1^: CD31 ^2^, CD3, CD79a
Lundgreen et al. [29]	1. Tenocytes 2. Ground substance 3. Collagen 4. Vascularity	1	n/a	Alcian Blue
Kurdziel et al. [31]	1. Tenocytes 2. Ground substance 3. Collagen 4. Vascularity	3	Random slides	Alcian Blue
Cook et al. [32]	1. Tenocytes 2. Ground substance 3. Collagen 4. Vascularity	n/a	n/a	Alcian Blue
Zabrzynski et al. [19]	1. Tenocytes 2. Ground substance 3. Collagen 4. Vascularity	3	n/a	Alcian Blue, Mallory, Masson Trichrome Inflammatory cells assesment
**Modifications in the Bonar Score**				
**Author**	**The Components of Bonar Score**	**Number of Investigators**	**Area of Specimen Investigation**	**Additional Staining**
Fearon et al. [21]	1. Tenocytes 2. Ground substance 3. Collagen 4. Vascularity 5. Cellularity	2	The most pathological area	n/a
Lundgreen et al. [27]	1. Ground substance 2. Collagen 3. Vascularity 4.Cellularity	1	n/a	Alcian Blue IHC: caspase 3, p53, KI67
Zabrzynski et al. [28]	1. Vascularity	n/a	The most pathological area	IHC: CD31, CD34
Lundgreen et al. [30]	1. Calcification 2. Morphology of adipocytes 3. Tenocytes 4. Ground substance 5. Collagen 6. Vascularity	2	n/a	Alcian Blue
Fearon et al. [35]	1. Tenocytes 2. Ground substance 3. Collagen 4. Vascularity 5. Cellularity 6. Adipocytes 7. Calcifiations	1	n/a	Substance P assessment Inflammatory cells assessment
Okazaki et al. [33]	1. Tenocytes 2. Ground substance 3. Collagen 4. Cellularity 5. Vascularity	2	n/a	IHC: collagen I type
Docking et al. [34]	1. Tenocytes 2. Ground substance 3. Collagen 4. Vascularity 5. Cellularity	3	n/a	Alcian Blue

^1^ Immunohistochemistry; ^2^ Cluster of differentiation.

**Table 4 medicina-57-00367-t004:** Clinical and radiological implications and Bonar score.

Author	Radiological Evaluation	Clinical Implications
Albano et al. [23]	MRI ^1^ (no correlation with Bonar; *p* = 0.937)	Clinical tests: single heel rise and first metatarsal rise (no correlation with Bonar score; *p* = 0.07)
Sethi et al. [24]	MRI, US ^2^ (no correlation with Bonar; respectively *p* = 0.08 and *p* = 0.368)	Clinical tests: ASES ^3^ Simple Shoulder Test, SF12 (no correlation with Bonar; *p* = 0.1301) Macroscopic assessment during arthroscopy (no correlation with Bonar *p* = 0.61, *p* = 0.42, *p* = 0.88)
Zabrzynski et al. [25]	US Damaged tissue regions corresponded to the areas identified during the sonographic and arthroscopic examinations)	n/a
Nuelle et al. [26]	MRI (MRI and intraoperative assessment did not show significant structural abnormalities within the tendon despite significant histopathologic changes)	n/a
Zabrzynski et al. [28]	n/a	VAS scale assessment, Tenderness over bicipital groove test (there was also no correlation between vessels ingrowth and pain; *p* = 0.2323)
Lundgreen et al. [27]	n/a	Bonar score of torn tendon demonstrated a significant degeneration compared with reference, intact samples (*p* < 0.005 and *p* < 0.001)
Lundgreen et al. [30]	n/a	The SST ^4^ from the smokers presented significantly more advanced degenerative changes (*p* < 0.001) The expression of protein p53 was also significantly stronger in the smokers (*p* = 0.024) Tenocyte density was significantly reduced in smokers compared with non-smokers (*p* = 0.019).
Kurdziel et al. [31]	n/a	No histologic differences in LHBT ^5^ specimens were observed in intact and torn RC ^6^ population.
Cook et al. [32]	US (US was normal in all but one of the 18 tendons having abnormal histopathology)	There were no differences between subjects with and without pathology in respect of training, recovery after surgery and basic anthropometric measures
Okazaki et al. [33]	n/a	Cleaning the hamstring tendons by scratching caused histological alterations and damage to type I collagen.
Docking et al. [34]	US, MRI (both imaging modalities demonstrated difficulty in identifying/differentiating between the presence of tendinosis, partial thick-ness tears, and full-thickness tears)	n/a
Fearon et al. [35]	n/a	Greater trochanteric pain syndrome group had a higher average Bonar score than control group (*p* = 0.04)

^1^ Ultrasounds; ^2^ Magnetic resonance imaging; ^3^ The American Shoulder and Elbow Surgeons Shoulder Score; ^4^ Supraspinatus tendon; ^5^ Long head of biceps tendon; ^6^ Rotator cuff tendons.

## Data Availability

Not applicable.

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
