# Peer review of "The Bonar Score in the Histopathological Assessment of Tendinopathy and Its Clinical Relevance—A Systematic Review"

_medicina, 2021, doi:10.3390/medicina57040367_

Round 1

Reviewer 1 Report

Dear author,

I appreciated the systematic review as it covers an important topic. Zabrzynska and colleagues performed a systematic review on the Bonar score's role in the histopathological assessment of tendinopathy. Although the manuscript is well written, I have some concerns. Therefore, I am not sure about the suitability of this paper for publication.

Abstract:

Too long. Please provide a shorter version.

Introduction

Line 47 – please explain better the differences in tendinopathy within different sports. Rotator cuff tendinopathy in thrower’s athletes presents different characteristics from knee tendinopathies.

Line 53 – too many references for the same sentence

Line 57 – add ref

Line 62 – awhich is a typo?

The authors should provide a valid reason for the clinical relevance of this study. Why should it be suitable for the publication? Why is it interesting for the international community?

Methods

Figure 1 – Why 37 articles have been excluded? Please add reasons. Why has 39 full text been excluded?

Results

Table 2 – add standard deviation to age and mean bonar. Each table should have a footnotes section with abbreviations.

Cook et al. include a young population as Okazaki et al. Young age could constitute a bias as a population are too heterogeneous. Please provide a more detailed explanation of this bias in the discussion.

The authors included only two articles on animal studies. If this paper aims to provide a correlation between clinical outcomes and Bonar score, I suggest deleting animal studies.

Line 164 – authors discuss Movin score, but this is not relevant for the study as authors focused on Bonar. I suggest removing the sentence.

I suggest reporting all the study limits in a Limitations section (Line 167 reports the bias due to the different number of investigators).

Authors should discuss better the difference between different types of tendons. Lesions are not the same in each site.

Table 4 – Zabrzynski et al (22) is it significant? Please add p value if present

                 Lundgreen et al (24) why bonar score is significant?

               Lundgreen et al (27) protein p53 expression is not a clinical implication. Tenocyte density is not a clinical implication

               Okazaki et al (31) the text in clinical relevance column is not relevant

Discussion

Some abbreviations are not mentioned before. Please add the extended text the first time you use an acronym. (eg. line 237, IHC, line 266 STG)

Line 272 – authros stated that bonar score had better agreement with clinics than with radiological modalities. Why do you conclude it? No statistical significance is reported in the radiological column of table 2. The majority of the studies reported not significant conclusions.

Line 292 – Lunedgree et al. (add ref. number)

Line 296 – Why did authors decide to discuss tobacco? Other comorbidities as diabetes or vasculopaties could damage the tendons.

The authors reported the conclusions without including animal studies. I suggest removing the two papers.

Authors should discuss if the bonar score change in the case of various sites (shoulder tendons are the same as knee tendons?)

Which is the level of evidence of the included studies? Please report and discuss

Why is this study relevant?

References

There are a lot of old references (older than 10-12 years). Please update the bibliography

Although the author cited some of their own paper that are relevant to the study, I suggest reducing the number of self-citations, using works published in a journal with a higher impact factor.

Author Response

Dear Editor,

Thank you for the opportunity to improve and resubmit our manuscript entitled:

“The Bonar score in the histopathological assessment of tendinopathy and its clinical relevance – a systematic review.”

The suggestions offered by the reviewers have been immensely helpful. We appreciate all the comments on the manuscript.

We have included the reviewer comments, and responded to them individually, indicating how we addressed each concern and describing the changes we have made. The revised manuscript has been read and approved by all the authors.

We wish to express again our appreciation for the insightful comments which have helped us significantly to improve our manuscript.

Yours sincerely,

Maria Zabrzyńska

Reviewer 1:

Overall, thank You for all the comments on the manuscript and good words.

Abstract: Too long. Please provide a shorter version.

Response: Thank you for the comment. We have extensively revised the abstract and a new version is much shorter.

Background and Objectives: This study was aimed to perform a comprehensive systematic review, which reports the role of the Bonar score in the histopathological assessment of tendinopathy and its clinical relevance. Materials and Methods: To identify all of the studies that reported relevant information on the Bonar scoring system and tendinopathy, an extensive search of the major and the most significant electronic databases (PubMed, Cochrane Central, ScienceDirect, SciELO, Web of Science) was performed. A systematic review of the literature was conducted according to the Preferred Reporting Items for Systematic Reviews and Meta-Analyses (PRISMA) guidelines. The extracted data included: year of study, geographical location, type of the study, radiological modifications, gender, number of patients, region of tendinopathy, mean age, control group, characteristics of the Bonar score and alterations in the scale, mean Bonar score, number of investigators, area of tendon investigation, clinical and radiological implications. Results: An extensive search of the databases and other sources yielded a total of 807 articles. The 18 papers were finally included in this systematic review and 13 original papers included the clinical and radiological implications of tendinopathy. Radiological evaluation was present in 8 studies (both MRI and US). The clinical implications were more frequent and present in 10 studies. Conclusions: Using the Bonar score, it is easy to quantify the pathological changes in tendinous tissue. However, its connection with clinics and radiological evaluation is much more complicated.  Based on the current state of knowledge, we concluded that the neovascularization variable in the Bonar system should be reconsidered. Ideally, the microscopic assessment score should follow the established classification scale with the radiological and clinical agreement and should have a prognostic value.

Introduction:

Line 47 – please explain better the differences in tendinopathy within different sports. Rotator cuff tendinopathy in thrower’s athletes presents different characteristics from knee tendinopathies.

Response: Thank you for the comment. We introduced some extra information about the association of tendinopathy and various sport activities.

“Chronic tendon pathology is the main problem in competitive and recreational athletes (tendon disorders lead to about 30% of orthopedic consultations. Achilles tendinopathy occurs most frequently in runners, basketball, volleyball and football players. On the other hand tibialis posterior tendinopathy is most common in runners. Hamstrings tendon disorders are often found in sprinters, jumpers and football players. Regarding to upper limb tendinopathies for arm extensors and flexors tendon disorders are usually present among tennis, golf players and throwing athletes, specifically javelin throwers and baseball players, as well as triceps tendon pathology. Rotator cuff and biceps tendinopathy are most commonly encountered in swimmers, American football players and javelin throwers.”

Line 53 – too many references for the same sentence

Response: Thank you for the comment, we fully agree with You. The references were revised.

Line 57 – add ref

Response: Thank you for the comment, we fully agree with You. The reference was added.

Line 62 – awhich is a typo?

Response: Thank you for the comment. The mistake was corrected. “Awhich” was replaced – “which”.

The authors should provide a valid reason for the clinical relevance of this study. Why should it be suitable for the publication? Why is it interesting for the international community?

Response: Thank you for the comment. This paper was invented, due the incoherence between the basic science and clinics in the tendon disorders field. We strongly believe that there is an association between the microscopic state of tendon and clinical manifestation of the disease. We aimed in this paper to show how the histopathological assessment of tendinopathy, used by numerous authors and in different tendons disorders, was linked with clinics, and moreover, if the histopathology has a reflection in clinics and treatment outcomes, what may be extremely important in further treatment planning.

Methods

Figure 1 – Why 37 articles have been excluded? Please add reasons. Why has 39 full text been excluded?

Response: Thank you for the comment. 37 papers were excluded after initial screening and abstract revision. Moreover, the next 39 papers were excluded after eligibility assessment: non–English language studies, case studies, reviews, letters to editors, conference abstracts, studies containing incomplete or irrelevant data and studies of level V evidence, any Bonar score application in structures other than the tendons (e.g., menisci) and studies in which the Bonar score was used for purposes other than pathology assessment.

Results

Table 2 – add standard deviation to age and mean bonar. Each table should have a footnotes section with abbreviations.

Response: Thank you for the comment. The text was modified according to your suggestion.

Cook et al. include a young population as Okazaki et al. Young age could constitute a bias as a population are too heterogeneous. Please provide a more detailed explanation of this bias in the discussion.

Response: Thank you for the comment. This kind of bias was discussed in the limitation section. “Okazaki et al. and Cook et al. presented a population with mean age lower than other authors and it could be connected with higher risk of bias.”

The authors included only two articles on animal studies. If this paper aims to provide a correlation between clinical outcomes and Bonar score, I suggest deleting animal studies.

Response: Thank you for the comment. According to Your and other Reviewer suggestion we excluded these two papers with animals.

Methods: The animal studies were also excluded.

Line 164 – authors discuss Movin score, but this is not relevant for the study as authors focused on Bonar. I suggest removing the sentence.

Response: Thank you for the comment. According to Your suggestion the text was deleted.

I suggest reporting all the study limits in a Limitations section (Line 167 reports the bias due to the different number of investigators).

Response: Thank you for the comment. This text was transferred to the limitation section.

Authors should discuss better the difference between different types of tendons. Lesions are not the same in each site.

Response: Thank you for the comment. We presented in the limitation section the higher risk of bias, due to the different types of tendons in present study. The tendons originated from shoulder predominated and histopathology may be different from lower limb. However, usually, the alterations in the course of tendinopathy in all tendons, of the upper and lower limbs, are usually similar and include disrupted collagen architecture, tenocyte morphological changes, neovascularization process, and expansion of the ground substance.

Table 4 – Zabrzynski et al (22) is it significant? Please add p value if present

                 Lundgreen et al (24) why bonar score is significant?

               Lundgreen et al (27) protein p53 expression is not a clinical implication. Tenocyte density is not a clinical implication

               Okazaki et al (31) the text in clinical relevance column is not relevant

Response: Thank you for the comment. None of the included sonographic modality studies presented a clear radiological association with histopathology. Authors showed no statistically significant correlation in their study (P>0.05). Lundgreen et al. revealed that the severely torn tendons had a higher Bonar score (P<0.05). Indeed, the protein p53 expression and tenocyte density is not a clinical implication, the impact of smoking on shoulder tendons pathology is extremely important, that is why we presented that smokers had a significantly elevated expression of protein p53 and the tenocyte density was highly reduced. In our opinion the mechanical stimuli and injury, presented by Okazaki et al., leads to severe histopathological alterations and should be included in clinical implications section.

Discussion

Some abbreviations are not mentioned before. Please add the extended text the first time you use an acronym. (eg. line 237, IHC, line 266 STG)

Response: Thank you for the comment. The manuscript was revised according to suggestion.

Line 272 – authros stated that bonar score had better agreement with clinics than with radiological modalities. Why do you conclude it? No statistical significance is reported in the radiological column of table 2. The majority of the studies reported not significant conclusions.

Response: Thank you for the comment. The manuscript was revised according to suggestion.

The papers with histopathological scores and clinical implication with significant statistical power: Lundgreen et al. reported that smokers have much more degenerated tendons of the shoulder as well as more extent apoptosis process (P<.001). Moreover, Sasaki et al. presented that scores were significantly higher in the denervated muscles and tendons (P<0.0001). Additionally, Fearon et al. showed that subjects with greater trochanter pain had higher Bonar scores (P=0.04)

We also revised the sentence: “These studies allow to conclude that the Bonar scoring system had better agreement with clinics than with radiological modalities.”

Into

“The previously mentioned studies suggest that the Bonar scoring system had better agreement with clinics than with radiological modalities, however this topic should be further explored.”

Line 292 – Lunedgree et al. (add ref. number)

 Response: Thank you for the comment. The references number was added.

Line 296 – Why did authors decide to discuss tobacco? Other comorbidities as diabetes or vasculopaties could damage the tendons.

Response: Thank you for the comment. Tobacco smoking seems to be strongly connected with neovascularization in tendinous tissue. The neovascularization is an important variable of the Bonar scoring system, and accented by numerous authors. On the other hand, present studies showed an ambiguous role of the new vessels ingrowth in tendinopathy process. Moreover, the neovascularization in other disciplines, such as ophthalmology, oncology and surgery may have a different impact on clinics. That is why the paragraph about the smoking and the neovascularization connection was introduced. We are also unable to analyze all of the comorbidities that may have an influence on the tendinopathy.

The authors reported the conclusions without including animal studies. I suggest removing the two papers.

Response: Thank you for the comment. According to Your and other Reviewer suggestion we excluded these two papers with animals.

Authors should discuss if the bonar score change in the case of various sites (shoulder tendons are the same as knee tendons?)

Response: Thank you for the comment. As it was written in the discussion; “The alterations in the course of tendinopathy in all tendons, of the upper and lower limbs, are usually similar and include disrupted collagen architecture, tenocyte morphological changes, neovascularization process, and expansion of the ground substance. The control group seems to be unnecessary in chronic tendinopathy due to well-known and established pathological pattern.”

We added the phrase: of the upper and lower limbs.

Which is the level of evidence of the included studies? Please report and discuss

Response: Thank you for the comment. The level of evidence was added to Table 2.

Why is this study relevant?

As we have mentioned before, the Bonar score application in various tendons pathology and also the modifications of the scoring system is relatively common. However, the link of histopathology with clinics and radiological assessment is yet not well explored. Our purpose was to perform a comprehensive systematic review, which reports the role of the Bonar score in the histopathological assessment of tendinopathy /upper and lower limbs/ and its association with clinics and radiological evaluation.

For example, during the arthroscopic treatment of the shoulder, usually, the surgeon does not harvest the damaged fragment of tendinous tissue, but in our opinion, the pathological alterations in tendon structure, may have an important impact on further treatment and rehabilitation, finally, on the clinical outcomes.

References

There are a lot of old references (older than 10-12 years). Please update the bibliography

Response: Thank you for the comment. The bibliography was updated.

Although the author cited some of their own paper that are relevant to the study, I suggest reducing the number of self-citations, using works published in a journal with a higher impact factor.

Response: Thank you for the comment. The studied published by Zabrzynski et al. are multicenter studies and despite the similar surname, the first author of these papers is someone else than the first author in the present study. All of the studies are from good quality, impact factor, journals. However, we enriched the present manuscript with a new, more actual references.

Reviewer 2:

Well written and performed study.

Unfortunately there are many limtations- as the authors claim themselves. Especially the heterogenity of papers is a proble as stated in the I am also concerned about clinical relevance as most of the studies don't focus on this matter.

I my opinion it should be considered to exclude animal studies, not sure if it#s really comparable.

Overall, thank You for the comments on the manuscript and good words. According to Your and other Reviewer suggestion we excluded these two papers with animals.

Reviewer 2 Report

Well written and performed study.

Unfortunately there are many limtations- as the authors claim themselves. Especially the heterogenity of papers is a proble as stated in the I am also concerned about clinical relevance as most of the studies don't focus on this matter.

I my opinion it should be considered to exclude animal studies, not sure if it#s really comparable.

Author Response

(The authors gave the same response as above.)

Round 2

Reviewer 1 Report

Authors modified the text according to the suggstions. I promote this paper for publishing in Medicina

Author Response

Dear Editor,

Thank you for your valuable suggestion.

Best regards,

Maria Zabrzyńska